# Integrated Purification and Formulation of an Active Pharmaceutical Ingredient via Agitated Bed Crystallization and Fluidized Bed Processing

**DOI:** 10.3390/pharmaceutics14051058

**Published:** 2022-05-14

**Authors:** Michael W. Stocker, Matthew J. Harding, Valerio Todaro, Anne Marie Healy, Steven Ferguson

**Affiliations:** 1School of Chemical and Bioprocess Engineering, University College Dublin, Belfield, D04 V1W8 Dublin, Ireland; michael.stocker@ucd.ie (M.W.S.); matthew.harding@ucd.ie (M.J.H.); 2I-Form, The SFI Research Centre for Advanced Manufacturing, School of Chemical and Bioprocess Engineering, University College Dublin, Belfield, D04 V1W8 Dublin, Ireland; 3SSPC, The SFI Research Centre for Pharmaceuticals, School of Pharmacy and Pharmaceutical Sciences, Panoz Institute, Trinity College Dublin, D02 PN40 Dublin, Ireland; todarov@tcd.ie (V.T.); healyam@tcd.ie (A.M.H.); 4SSPC, The SFI Research Centre for Pharmaceuticals, School of Chemical and Bioprocess Engineering, University College Dublin, Belfield, D04 V1W8 Dublin, Ireland; 5National Institute for Bioprocess Research and Training, 24 Foster Avenue, Blackrock, Co., Belfield, A94 X099 Dublin, Ireland

**Keywords:** co-processing, crystallization, fluidized bed coating, formulation, isolation-free manufacturing, pharmaceuticals

## Abstract

Integrated API and drug product processing enable molecules with high clinical efficacy but poor physicochemical characteristics to be commercialized by direct co-processing with excipients to produce advanced multicomponent intermediates. Furthermore, developing isolation-free frameworks would enable end-to-end continuous processing of drugs. The aim of this work was to purify a model API (sodium ibuprofen) and impurity (ibuprofen ethyl ester) system and then directly process it into a solid-state formulation without isolating a solid API phase. Confined agitated bed crystallization is proposed to purify a liquid stream of impure API from 4% to 0.2% *w*/*w* impurity content through periodic or parallelized operations. This stream is combined with a polymer solution in an intermediary tank, enabling the API to be spray coated directly onto microcrystalline cellulose beads. The spray coating process was developed using a Design of Experiments approach, allowing control over the drug loading efficiency and the crystallinity of the API on the beads by altering the process parameters. The DoE study indicated that the solvent volume was the dominant factor controlling the drug loading efficiency, while a combination of factors influenced the crystallinity. The products from the fluidized bed are ideal for processing into final drug products and can subsequently be coated to control drug release.

## 1. Introduction

Crystallization is the most widely used purification and separation process for the large-scale production of small molecule active pharmaceutical ingredients (APIs). A crystallization process typically used for API manufacturing achieves high product purity and successful isolation of the desired polymorphic form while meeting requirements for high yield and productivity [1,2]. As most industrial crystallizations are batch processes, productivity is often limited and has, in part, led to substantial research effort in developing continuous crystallization processes [3,4,5,6,7]. In addition to improved productivity, continuous crystallization has a number of distinct advantages over batch operation. Being able to reach a steady state during the crystallization process (something not possible in batch) results in greater reproducibility and consistency in the crystalline material and shows some advantages compared to batch in terms of control of polymorphic form and engineering of particle size distribution, in addition to a significant increase in productivity [8,9,10]. In addition, competitive steady states in continuous crystallizers have recently been demonstrated to provide a robust mechanism to produce both stable and metastable polymorphs at scale [11,12,13,14]. In many cases, this mitigates the need for downstream processing of the powder, which in turn reduces production costs and eliminates the risk of solid state transformations during these operations [15].

Despite the advantages that continuous crystallization presents, it remains one of the most challenging operations to conduct on the relatively small scale associated with continuous pharmaceutical production. Intermittent and pneumatic withdrawal and slurry withdrawal methodologies have enabled sustained operation for continuous stirred tank crystallizers [16,17,18]. However, in addition to the need to operate with high solids loadings and handle supersaturated slurries that have an inherent tendency to foul reactor surfaces, filtration and, in some cases, drying and accurate reconstitution (if used for intermediate purification or solvent adjustment) must also be conducted and can also present significant challenges. For this reason, solids are, for the most part, avoided in flow chemistry and continuous synthetic route development and so, for improved process integration, it is advantageous for continuous crystallization operations to also avoid solid effluents in order to avoid blockages and fouling of process lines [6,19,20]. This designed telescoping of steps together in flow chemistry and continuous manufacturing has led to primarily using liquid-liquid extraction for purification purposes, excluding the high level of intermediate purification and unincumbered solvent selection that crystallization supports in batch synthesis of pharmaceuticals.

Recent advances have managed to avoid the presence of solids by using semi-continuous falling film crystallization and re-dissolving the purified solid product in fresh solvent [21,22]. The development of this and similar approaches will allow the high selectivity, high productivity and low cost provided by crystallization operations to be realized and applied to a more diverse range of continuous processes [3]. In this paper, a new semi-continuous confined suspension crystallization approach is deployed in a similar manner to these semi-continuous falling film solution layer crystallizations. While this does not completely eliminate the isolation of particulates as in layer crystallization, it does eliminate the need to transfer or handle slurries. On a single-stage basis, suspension crystallizations have been shown to have a higher purification efficiency than layer crystallization in melt crystallization applications [23] and this is likely the case for solution layer crystallization also. In the case of this study, this separation represents the final opportunity to purge impurities before the formation of the drug product, so confined suspension crystallization was preferred.

Fluidized bed coating is a versatile process widely used across pharmaceutical industries to improve the physical characteristics of solid products. Coating processes have found applications in taste masking, improving product flowability, moisture protection of final dosage forms and controlling the release profile of APIs [24,25,26,27]. In fluidized bed coating, the particles to be coated are suspended in a continuous flow of air or gas, such as nitrogen, and passed through an atomized stream of coating material until the desired degree of coating has been achieved.

Of the main coating processes used for coating solid dosage forms, the bottom spray Wurster process is generally regarded as the most efficient for producing a high-quality product [28]. What distinguishes Wurster coatings from other spray coating processes is the nature of the flow of the substrate to be coated within the fluidized bed. In contrast to the random and almost static movement of the product found in top-sprayed processes, the product in the Wurster process is cycled through the coating stream in a controlled manner that resembles a waterspout [29]. The lowest chamber of the rig is equipped with a hollow cylindrical partition (the Wurster insert) that is slightly elevated from the bottomplate of the bed. The atomizer is positioned beneath the center of the insert, which forms a shroud around the atomized coating solution. The bottomplate is designed to have the greatest air flow in the center of the rig, which, in combination with the Wurster insert, results in two different air velocity zones in the lower section of the fluidized bed and forces the substrate to cycle through the partition at high velocity and contact the coating stream. The well-defined circulating flow of the substrate gives rise to the increased quality and efficiency of the process [30].

Coating inert starter cores/beads with APIs has already been shown to be a viable formulation approach for crystalline drugs and their salts [31], thermally sensitive amorphous solid dispersions [32] and cocrystals [33] and remains of interest for drugs currently in development [34]. Additionally, multiple layers can be built up on the surface of the bead after the initial API layer, creating further opportunities for tuning the properties of the product [35]. This allows all the desired attributes of the final formulation to be built into each pellet, allowing for an appropriate dose to be dispensed directly into empty hard capsules for packaging and distribution, and the same product can potentially be tableted by direct compression.

While alternative examples of in situ heterogeneous crystallization of API with excipients have been reported, these processes still require a number of steps between the final purification step and the finished final dosage form [36,37]. After crystallization, the API requires filtration and drying before being processed and combined with other excipients in the formulation/drug product plant, where it commonly undergoes a combination of some or all of: milling, blending, granulation, compression and coating. Directly tableting an API (acetaminophen) that was crystallized directly onto the surface of an excipient (D-mannitol) using different continuous heterogeneous crystallization processes has been demonstrated as a route to mitigating the majority of the additional steps required in the production of a final dosage form. Despite requiring filtration only between purification and formulation steps, the tablets produced from the product had low hardness, which might be improved by using a different excipient at the expense of the performance of the crystallization [38]. Alternatively, model APIs have been crystallized directly into microparticles with sucrose shells that can encapsulate hydrophobic and hydrophilic APIs [39]. This was achieved using microfluidic crystallization techniques that enabled crystallization and co-formulation to take place simultaneously and were successfully adapted to run continuously [40].

The liquid effluent from the aforementioned confined suspension crystallization process employed in the current study allows the process to be combined with the spray coater as a continuous purification–formulation route. The process intensification and elimination of solid handling that results from this approach relative to conventional routes is shown in Figure 1. While the fluidized bed apparatus used for the current study is not suitable for fully continuous operation, as it requires the fluidized bed to be disassembled for the solid product to be recovered, the crystallization operation can be used in continuous operation with the addition of a surge tank, as was conducted here. For high volume drug scale-up, parallelized purification and intermittent or fully continuous operation can be utilized with Wurster coating and fluidized bed spray coating more generally with production scale machinery [41,42].

In this work, the objective was to use a Design of Experiments (DoE) approach to develop a spray coating process that allows for the drug loading efficiency and the degree of crystalline material on the product to be tightly controlled, while simultaneously intensifying the manufacturing process. The spray coating process was fed with the product from a confined suspension-agitated bed suspension crystallization process, as per the process scheme outlined in Figure 2. This results in substantial process intensification, as purification and formulation are performed in two unit operations that can be adapted to run semi-continuously and integrated with existing continuous manufacturing processes.

## 2. Materials and Methods

### 2.1. Materials

Sodium ibuprofen (Na Ibu) was purchased from Santa Cruz Biotechnnology (Santa Cruz, CA, USA) with a purity > 99% and free acid ibuprofen was purchased from Kemprotec Limited (Carnforth, UK). Microcrystalline cellulose (MCC) beads (Cellets^®^) were obtained from Pharmatrans Sanaq AG (Allschwi, Switzerland). The methanol and ethanol used for the coating process were supplied by Corcoran Chemicals (Dublin, Ireland). Polyvinylpyrrolidone (PVP K-25, M_w_ 31,000 g mol^−1^) was supplied by BASF (Ludwigshafen, Germany) and hydroxypropyl methylcellulose (HPMC, Pharmacoat^®^ 606, M_w_ 32,800 g mol^−1^) was donated by Shin-Etsu Chemical Co. (Tokyo, Japan). HPLC-grade methanol and acetonitrile were purchased from Fisher Scientific (Dublin, Ireland). Monobasic potassium phosphate and sodium hydroxide for buffer preparation were purchased from Sigma Aldrich (Wicklow, Ireland) and Fisher Scientific (Dublin, Ireland), respectively. The ibuprofen ethyl ester impurity (91% purity) was prepared via Fischer–Speier esterification, as previously described [22].

### 2.2. Agitated Bed Crystallizer

In the agitated bed suspension crystallizer, seed material is suspended within a glass column and a physical stirrer (either overhead impeller or magnetic stir bar) is used to keep the material suspended as crystal growth occurs. Figure 3 shows the design of the confined suspension agitated bed crystallizer and the process flow diagram. Customized polytetrafluoroethylene (PTFE) caps at the base and top of the vessel allow for the addition of other components. The base PTFE cap houses a filter disc (25 mm Ø, 3 mm thickness), which retains the crystal bed during washing and dissolution steps. The top PTFE cap has several ports to allow the insertion of process analytical technology (PAT) or an overhead stirrer.

The operation of the crystallizer is as follows: A dose of a supersaturated solution is fed into a cooled and suspended seed bed, resulting in rapid growth of crystals. Once the crystal growth is complete, the column is drained through an integrated filter disc and the retained crystals are washed. After a second drain process, pure solvent is added to re-dissolve the product in the desired solvent, thereby integrating a possible solvent swap into the purification step and allowing for concentration adjustment. Combined with other similar units run in tandem, the process can be run as a semi-continuous unit operation.

### 2.3. Design of Experiments (DoE)

A five-factor, two-level fractional factorial study was produced using Design Expert 10 software (Stat-Ease Inc., Minneapolis, MN, USA) to screen for the parameters with the most significant influence on the spray coating process. From preliminary experiments, certain parameters were found to be sensitive to other inputs to maintain fluidization and so were kept constant throughout the study:Solution feed rate: 2.1 mL min^−1^Fluidizing air flow: 30 m^3^ h^−1^Atomization pressure: 2 bar

Of the five factors selected for investigation, three were numerical and two were categorical:A. Inlet air temperature: 50 or 60 °CB. Binder type: PVP or HPMCC. Binder mass: 1 or 3 gD. Solvent volume: 50 or 100 mLE. Solvent: methanol or ethanol

For each run, 10 g of API was sprayed onto 88 g of MCC beads to give an average total batch size of 100 g. 16 runs were generated using the DoE software, the conditions of which are detailed in Table 1. All solutions were prepared in the same manner, where the API was first dissolved in the desired solvent before the binder was added. The beads were coated using a Mini-Glatt (Glatt, Binzen, Germany) fluidized bed equipped with a Wurster insert and a 0.5 mm spray nozzle diameter. After the solution was sprayed, the coated beads were dried in the fluidized bed for 30 min.

### 2.4. Powder X-ray Diffraction (pXRD)

Materials were analyzed using a Miniflex II X-ray diffractometer (Rigaku, Neu-Isenburg, Germany) with Ni-filtered CuK α radiation (1.54 Å). The tube voltage and current were 30 kV and 25 mA, respectively. In the case of coated and uncoated MCC beads, these were lightly ground in a pestle and mortar prior to analysis and all samples were analyzed on a zero background silicon sample holder in reflection mode. Diffraction patterns were obtained for 2θ between 2 and 40° at a step scan rate of 0.05° per second.

### 2.5. Drug Loading Efficiency (DLE)

The drug loading efficiency (DLE) is analogous to the yield and is the proportion of API in the solution that is successfully loaded onto the beads. Due to the high ratio of inert starter core/beads to API, gravimetric yields are significantly skewed by a loss of even a small amount of bead to the filters or joints in the fluidized bed or elsewhere during the recovery of the product. DLE was determined by placing 1 g of beads in an excess of a suitable solvent (methanol) and leaching the loaded API into the solution. The concentration of drug in the sample was quantified by high-performance liquid chromatography (HPLC) using a Waters 2695 separations module system (Milford, MA, USA) equipped with a Waters 2996 photodiode array detector (Milford, MA, USA) and detection wavelength of 220 nm. The method was run isocratically at 40 °C using a Phenomenex C18 column (150 × 4.6 mm, 5 µm) with an injection volume of 10 µL. The mobile phase was a 50/50 (*v*/*v*) mixture of acetonitrile and phosphate buffer (pH 7.2) at a flow rate of 3 mL min^−1^. The limit of quantification was 8.9 µg mL^−1^.

### 2.6. Degree of Crystallinity (DoC)

To quantify the amount of crystalline drug on the surface of the beads, the results from the DLE were used in combination with the heat of fusion, as quantified by modulated differential scanning calorimetry (DSC) using a method previously described by Serrano et al. [33]. In short, the heat of fusion of the API on the bead was calculated as a percentage of the heat of fusion of the same amount of pure crystalline drug relative to a literature value [43], as shown in Equation (1). DSC was carried out using a TA Q200 instrument (TA Instruments, Elstree, UK) under an inert atmosphere. Between 3–4 mg of coated beads were placed in sealed standard aluminum pans and heated at a rate of 5 °C min^−1^ with a modulation of ±0.796 °C min^−1^ from 0 to 250 °C.
(1)$$\mathrm{DoC}=100\cdot \left(\frac{\Delta H_{f,\;\;\;API\;on\;beads}}{\left(DLE\cdot 0.01\right)\cdot \Delta H_{f,\;\;\;Crystalline\;API}}\right)$$

## 3. Results and Discussion

### 3.1. Fluidised Bed Coating Process Development

The initial studies identified two key process limitations that were considered when designing the study. First, the amount of material that can be fluidized is limited by the available supply of nitrogen. While the fluidized bed is rated for up to 50 m^3^ h^−1^, the available supply in the house is 32 m^3^ h^−1^ and so, in order to mitigate fluctuations in supply, a stable maximum fluidizing air flow of 30 m^3^ h^−1^ was used for all runs. Second, it was found to be unfeasible to use the free acid form of ibuprofen. During runs probing extremes of temperature, concentration, binder mass and use of lubricant (talc), fluidization ceased after a variable amount of time. Initially, this was thought to be due to the system being too wet as a result of insufficient removal of the solvent from the solid phase; however, further investigation using low concentration and higher temperature showed the problem to be unrelated to the processing parameters.

The underlying cause for the run failure was found to be the challenging material properties of ibuprofen. Literature indicates the melting and glass transition temperatures of ibuprofen to be 76 °C and −45 °C respectively [44] and so, during the runs at the high temperature extreme (80 °C), the drug being in the liquid state is what led to the loss of fluidization. The low glass transition temperature of the API means that the material remains cohesive during runs at lower temperatures. Once enough API has been applied to the beads, this results in agglomeration and the loss of fluidization and subsequent failure of the run. To mitigate this issue, the sodium salt of ibuprofen (Na Ibu) was used, as it has a significantly higher melting point (200 °C) and glass transition temperature (78 °C) [45]. HPMC and PVP were selected as binders as they have previously been used successfully as binders for granulating ibuprofen and MCC in a fluidized bed process, so it was anticipated that they would be able to perform a similar function during the coating process [46]. It should be noted that HPMC (binder) is ordinarily insoluble in anhydrous alcohol. However, when added to a solution of ethanol or methanol containing Na Ibu, HPMC was readily dispersed in the solvent. This is attributed to Na Ibu acting as a surfactant between the polymer and the solvent to stabilize dispersion. The structures of the components of the spray coating solution are shown in Figure 4.

#### 3.1.1. DoE Characterization Study

DoE is a statistical tool that involves planning, executing and analyzing a set of experiments in such a way that valid conclusions can be drawn from a relatively small dataset. Such studies allow multiple process variables to be simultaneously investigated efficiently for their effect on the overall system [47]. As the application of layers of solid material to the surface of a substrate is a complex process with a multitude of variables interacting at any one time, DoE is well suited to developing a robust, high-quality process [33,48]. Coating processes described in the literature have almost exclusively been developed on similar machines using a significantly smaller batch size (between 5 and 25 g) [26,33] or on larger scale machines operating on a kilogram scale [30,49,50], and so are, for the most part, incomparable to the intended 100 g batch size of this work. In order to minimize the amount of material consumed and the time spent on developing a robust process suitable for use in conjunction with the new crystallization process, a DoE approach was taken. Preliminary work showed that the system was most sensitive to specific parameters when trying to maintain fluidization, namely solution feed rate, fluidizing air flow and atomization pressure. The results of the DoE study are shown in Table 2. The DLE was determined by HPLC and the DoC was calculated from the heat of fusion of the API on the beads relative to that of the purely crystalline API and the DLE.

DoE allows the most statistically influential factors to be quickly and easily identified. The Pareto charts in Figure 5 show the results of the statistical model. The *y*-axis corresponds to the magnitude of the effect that changing the factor(s) has on the selected response. All five factors were selected for analysis, as well as all combinations of factors deemed statistically significant, i.e., the magnitude of the effect was above the t-value limit. Orange and blue correspond to positive and negative effects, respectively, while the factors are as follows: A is the inlet air temperature, B is the binder type, C is the binder mass, D is the solvent volume, and E is the solvent type. White bars superimposed on the solid colored bars show the factors that were selected for inclusion in the experimental model.

The resulting models had *p*-values of 0.0117 for the DLE and 0.0016 for the DoC, and R-squared values of 0.9831 and 0.9905, respectively. From the Pareto charts in Figure 5, the most influential factor by far for the DLE is the solvent volume, followed by the binary interactions between the binder type and solvent volume, and the binder mass and solvent volume. With regard to the DoC, the most significant factor was the binary interaction between the inlet air temperature and the solvent volume, followed by the binder mass and the binder type. The equations for DLE and DoC in terms of the coded factors are given in Equations (2) and (3) and the final equations in terms of the actual factors can be found in the Appendix A.
(2)$$\begin{array}{cc} \mathrm{DLE}\;=76.79+ & \left(0.09\cdot A\right)+\left(0.07\cdot B\right)+\left(2.32\cdot C\right)+\left(8.57\cdot D\right)+\left(3.12\cdot E\right)+\left(2.56\cdot A\cdot D\right)\\ & -\left(2.71\cdot A\cdot E\right)+\left(3.69\cdot B\cdot D\right)-\left(4.46\cdot B\cdot E\right)+\left(2.64\cdot C\cdot D\right)+\left(3.99\cdot C\cdot E\right) \end{array}$$
(3)$$\begin{array}{cc} \mathrm{DoC}\;=43.79- & \left(4.47\cdot A\right)-\left(8.07\cdot B\right)-\left(9.42\cdot C\right)-\left(6.45\cdot D\right)-\left(4.61\cdot E\right)+\left(9.66\cdot A\cdot E\right)\\ & +\left(7.93\cdot B\cdot C\right)-\left(5.03\cdot B\cdot D\right)+\left(5.03\cdot B\cdot E\right)+\left(4.51\cdot C\cdot D\right)+\left(7.83\cdot C\cdot E\right) \end{array}$$

#### 3.1.2. pXRD

pXRD was used to verify the presence of crystalline material and quantify the DoC of the API. However, no useful information can be discerned from the diffractograms, as the crystalline Na Ibu peaks are overshadowed by the pattern from the amorphous MCC cores, with only a small peak visible at 3.75 degrees (2θ). This can be seen in Figure 6, which shows the diffractogram for the products from Run 1 compared to the diffractogram for Na Ibu and uncoated MCC beads (similar diffractograms were obtained for the products of all runs). While it was not possible to quantify the amount of crystalline material on the beads using this method, the presence of this peak confirms that the API was successfully loaded onto and crystallized on the surface of the beads. Additionally, it should be noted that while it might theoretically be possible that the polymorphic form of the API on the beads is that of the less stable monotropes of the α and β forms of the API, the consistent presence of the characteristic peak at 3.75 degrees (2θ) of the stable γ form (as opposed to the characteristic peaks of the α and β forms that are expected at 5.2 and 5.7 degrees, and 5.9 degrees 2θ, respectively) strongly indicates that this is the form of the API on the beads [51].

#### 3.1.3. Drug Loading Efficiency

The contour plots presented in Figure 7 demonstrate the interactions between binder type solvent type, temperature and solvent volume when 1 g of binder is used. In all cases, methanol gave a better DLE than ethanol. This is attributed to the lower boiling point of methanol, resulting in a more rapid viscosity increase after the coating solution contacts the beads [52]. The increase in viscosity causes the solution to become more cohesive, so more of the coating adheres to the bead surface with each cycle. Higher loading is also achieved when a greater volume of solvent is used, which can be explained by the solution still being wet when it reaches the beads [53]. If the solution is devoid of solvent by the time it comes into contact with the beads, then it is essentially being spray dried and is thus unable to adhere to the substrate and form layers. The same rationale can be applied to the effects of temperature on the system; higher temperatures result in a greater proportion of the material being spray dried and so the DLE is again reduced [54].

The effect of temperature on the system was reversed when ethanol was used instead of methanol. The results show that increasing the temperature results in an increase in loading when the solvent was ethanol, whereas the loading for methanol improved at lower temperatures. Owing to differences in boiling point, droplets of ethanolic solutions are unable to dry quickly enough to adhere to the surface at lower temperatures. However, when this is countered by using a low solvent volume, the droplets dry too quickly in the lower part of the bed, where the velocity of the drying air is greatest and so still require a high solvent volume to ensure the droplets are wet enough when coming into contact with the substrate. In general, temperature had a more profound influence when methanol was used. This is likely due to the temperature range studied being much closer to the boiling point of methanol than ethanol, making the system more sensitive to fluctuations. Of the two binders, PVP gave the best results overall but was outperformed by HPMC in ethanol. However, the difference in the DLE of HPMC in either solvent is only slight and so variations are probably the result of factors not covered in this study.

The trends shown in Figure 8 for when 3 g of binder was used are largely the same as for when 1 g of binder was used with regard to the process conditions; high solvent volume and low temperature give the highest loading for methanol, while high volume and high temperature gave the best results for ethanol. When methanol was used as the solvent, the temperature and solvent volume had a slightly lessened effect than was the case for 1 g of binder and resulted in slightly lower loading. This can again be attributed to methanol’s low boiling point being relatively close to the temperature range investigated and drying at a similar rate to that when a lower amount of binder was used. This means that there is less time during the coating cycle for the effect of the additional binder to take effect. For the ethanolic solutions, the drying rate of the solution once it comes into contact with the MCC beads is more prolonged, so the loading is significantly increased with a higher binder mass. In contrast to when 1 g was used, the best DLE that was achieved out of all the systems that were studied was when HPMC was used in conjunction with ethanol. This could be due to it having a higher initial solution viscosity, which allows more of the atomized feed to adhere to the beads on each cycle.

#### 3.1.4. Degree of Crystallinity

From the data presented in Figure 9 and Figure 10, the most favorable conditions for a high DoC are similar when 1 g and 3 g of binder are used. In both instances, ethanol yielded a significantly higher DoC than methanol. As was the case with the drug loading, this is the result of differences in the boiling points of the two solvents: ethanol, with its higher boiling point, dries more slowly than methanol, so there is more time for crystallization to occur and so the proportion of crystalline material is increased. This is also reflected in the effect of temperature on the system; at lower temperatures, drying occurs at a slower rate, allowing more time for crystallization, which in turn results in a higher DoC. The faster rate at which methanol dries causes the DoC to be much lower. The size of this effect is large enough that it cannot be countered by varying the other parameters to increase the DoC, even when such a low binder mass is used. This effect may be offset by the lower solubility of the API in ethanol compared to methanol [55], which may increase the relative rate of supersaturation generation at the same rate of evaporation.

When 3 g of binder was used, the DoC was significantly lower in all cases than when 1 g of binder was used. This can be explained by the additional polymer increasing the distance between the API molecules, making it harder for the molecules to aggregate to the critical concentration for nucleation to occur. The presence of extra polymer in the solution also means that during each cycle through the fluidized bed, there is a lower chance that the API in the droplets will come into contact with crystalline material already on the beads, acting as sites for crystal growth. In addition, the greater amount of polymer results in an increase in solution viscosity as the droplets are drying on the surface of the beads, which in turn causes the diffusivity of the drug in solution to drop [56]. As crystallization is proportional to diffusivity, the rate of growth during each cycle is therefore reduced, again resulting in a decrease in the degree of crystallization. From the Pareto chart in Figure 5, it can be seen that the DoC is significantly influenced by the binder mass, which causes the relationships between DoC and other process variables to be masked by the dominant inverse relationship between DoC and increasing binder mass.

Changing the solvent volume has a similar relationship to binder mass when ethanol is the solvent: lower volumes mean that the solution is more concentrated and thus more favorable for crystallization to occur. However, at low temperatures when using 1 g of binder, this effect becomes inconsequential as the DoC approaches 100%. In contrast to the ethanolic systems, where methanol was the solvent, a higher inlet air temperature resulted in a higher DoC. This is surprising, as it would be assumed that as the system approaches the boiling point of the solvent (as is the case here), the time available for crystallization to take place is reduced and would be expected to result in a lower DoC. In comparison to PVP, HPMC has a significantly larger number of functional groups that can form hydrogen bonds with Na Ibu molecules, as shown in Figure 4. As hydrogen bonding represents an additional energy barrier that must be overcome before nucleation can occur, it would be expected that there would be some differences between the effects of using the two polymers [57]. However, as there is no significant variation between the two sets of results, this suggests that the effects of differences in hydrogen bonding between the polymer and drug molecules that arise from structural differences are negligible.

### 3.2. Combining Crystallisation and Spray Coating Operations

An agitated bed suspension crystallization process was used to generate the purified API solutions for the combined runs. For this process, the feed solution was a 10% *w*/*w* solution of Na Ibu in 90:10 n-heptane:ethanol with 4% *w*/*w* impurity heated to 50 °C. The impurity used in this work is the ethyl ester of ibuprofen, which was identified as a good test system owing to its similar structure to the desired product. The crystallizer initially contained 50 mL of the saturated 90:10 n-heptane:ethanol solvent system at 5 °C (acting as the suspension solvent for the feed) and 5% *w*/*w* seed crystals relative to the mass of Na Ibu in the feed solution. After rapidly dosing the feed into the agitated bed, crystal growth was allowed to proceed over 1 h before the mother liquor was drained and the purified product was retained in the bed. 50 mL of pure heptane was added to the bed and the Na Ibu resuspended in order to remove any residual impurity in the crystal matrix and was then drained from the vessel. Finally, the bed was heated to 20 °C and 50 mL of the solvent required for the following process (methanol or ethanol) was dosed into it to redissolve the product, allowing it to be recovered. The product from the crystallization contained Na Ibu at 80% yield dissolved in approximately 50 mL of the recovery solvent and contained less than a 0.2% impurity. For the combined runs with the fluidized bed, 14 g of Na Ibu was used in the crystallization feed solution in order to ensure that the productivity of the process matched that of the fluidized bed, after taking into account the process yield. The required volume of solvent for dilution and the mass of the polymer binder was calculated on a g/g basis. The purity of the product was determined using the HPLC method previously described [22].

Based on the results from the DoE study shown in Table 2 and the model equations generated from the data (Equations (2) and (3)), run conditions were selected to demonstrate optimization of the process with different goals—maximization of DLE and maximization of DoC. Effluents based on the run conditions were produced by re-dissolving the API in pure solvent before dilution with an appropriate polymer solution, which was fed into the surge tank via a peristaltic pump, as shown in Figure 2. This allowed the HPMC to be easily transferred as a slurry to the vessel containing the Na Ibu, whereupon it formed a stable dispersion. The mass of beads used was kept the same as that used during the DoE study, as were the other process parameters which were held constant in the DoE study (atomizing air pressure, fluidizing air flow and solution feed rate).

After carrying out the runs detailed in Table 3, the beads were analyzed via the methodologies previously described and the results are summarized in Table 4, along with the predicted values obtained from numerical optimization in the Design Expert 10 software using the models developed from the DoE study. The DLE for Run (i) is predicted to be in excess of 100%, which is due to the inability to set maximum attainable values in the model. In spite of this, the experimental results are in good agreement with the predicted value. The slight decrease is attributed to the small amount of residual heptane in the feed solution, which reduces the dispersibility of the HPMC binder, thereby diminishing how well it can stick the API to the surface of the beads. This effect appears to have been even more significant in Run (ii) when PVP was used as the binder, where the actual DLE is more than 25% lower than predicted. Despite the smaller mass of binder being used, the reduced solubility in this case results in the PVP drying more quickly and so remaining cohesive for a shorter amount of time.

The predicted DoC in Run (i) is in good agreement with the predicted value and the presence of heptane appears to result in only a small decrease in the experimental DoC. In Run (ii), where the aim was to increase the DoC, the predicted value was marginally higher than the experimental one. This is again attributed to the presence of heptane, which forces the Na Ibu to precipitate from the solution more quickly than when pure ethanol is used as the solvent, reducing the degree to which the API is able to crystallize. The lower-than-expected loading may also contribute to this, as there are fewer active sites on the beads to induce crystallization as the droplets dry.

## 4. Conclusions

A novel isolation-free confined suspension crystallization process has been successfully integrated with a fluidized bed coater to enable the isolation-free manufacturing of a solid-state formulation of a model API Sodium Ibuprofen. Relationships between process variables and process outcomes were identified and used to select desired feed compositions and process conditions for runs to maximize both drug loading efficiency onto coated inert cores/beads and the degree of crystallinity of the coated API. This approach enables purification and formulation to be efficiently carried out in two-unit operations. Although carried out on a relatively small scale, all aspects of the process are readily scalable with suitable equipment and therefore represent a platform for semi-continuous purification to formulation to be carried out in two combined unit operations with the capability to take material continuously from continuous upstream drug substance production if as needed. There is also potential to further develop the fluidized bed processing aspect of the manufacturing approach to enable additional coating of API-coated beads in order to facilitate controlled or modified drug release.

## Figures and Tables

**Figure 1 pharmaceutics-14-01058-f001:**
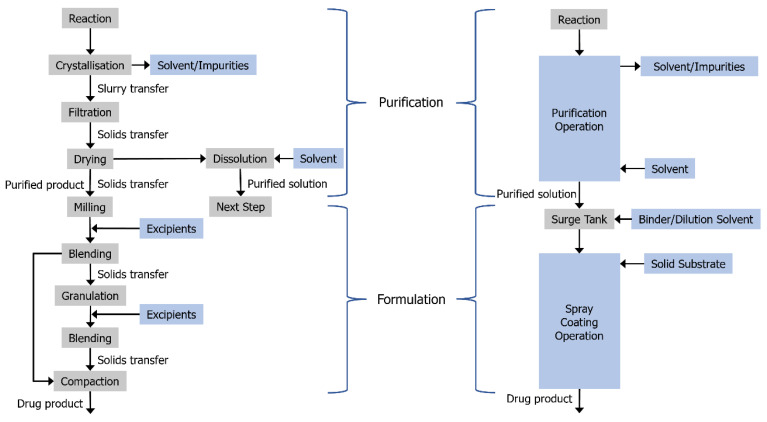
Schematic representation of the slurry- and powder-free manufacturing scheme afforded by utilizing isolation-free manufacturing techniques. Adapted with permission from Ref. [21]. 2016, American Chemical Society.

**Figure 2 pharmaceutics-14-01058-f002:**
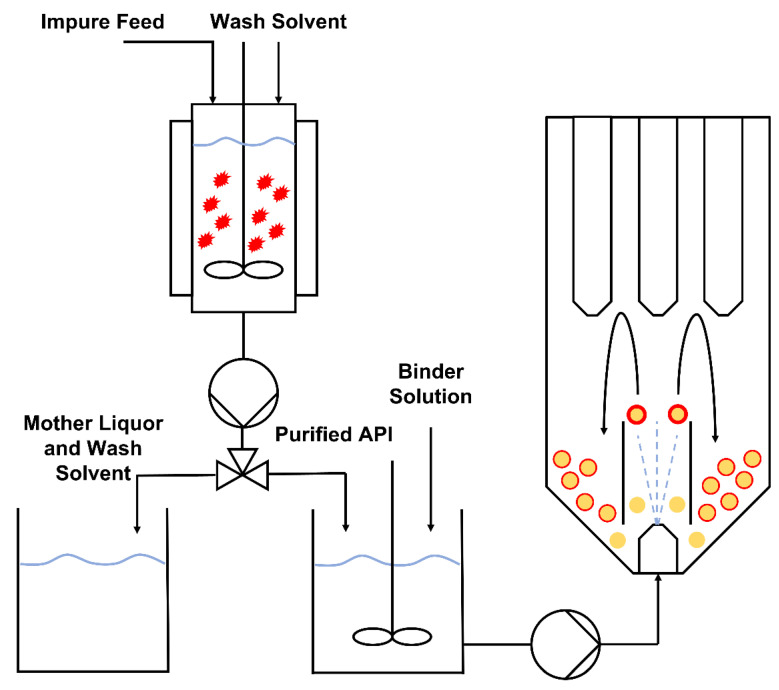
Process flow diagram of the combined crystallization and fluidized bed-coating processes.

**Figure 3 pharmaceutics-14-01058-f003:**
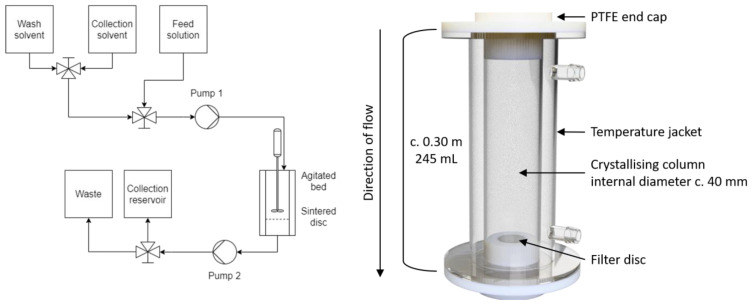
Diagrams of the agitated bed crystallizer. (**Left**) Process flow diagram of the agitated bed crystallizer, (**Right**) Design of the Confined suspension–agitated bed crystallizer.

**Figure 4 pharmaceutics-14-01058-f004:**
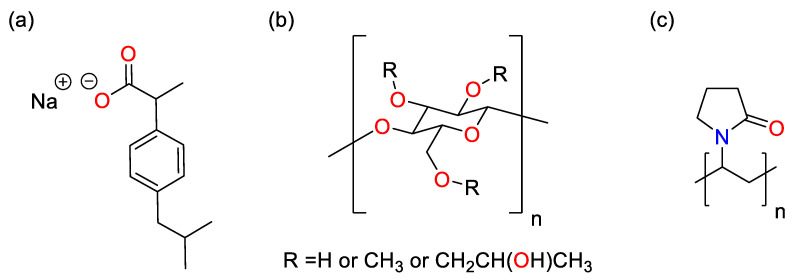
Structures of components used in the spray coating process. (**a**) Na Ibu, (**b**) HPMC, and (**c**) PVP.

**Figure 5 pharmaceutics-14-01058-f005:**
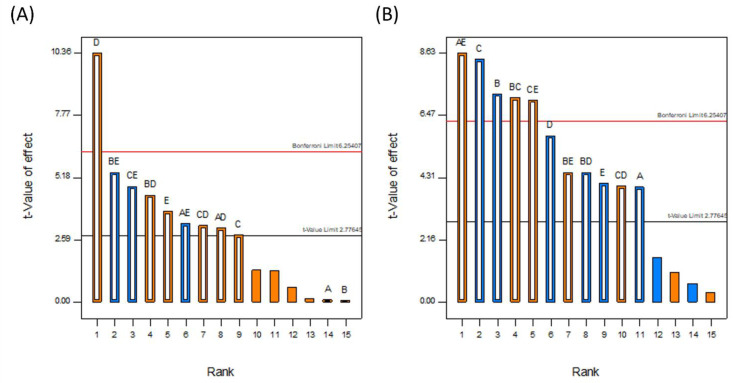
Pareto charts showing the rank and magnitude of the factors and combinations of factors considered in the study for (**A**) DLE and (**B**) DoC.

**Figure 6 pharmaceutics-14-01058-f006:**
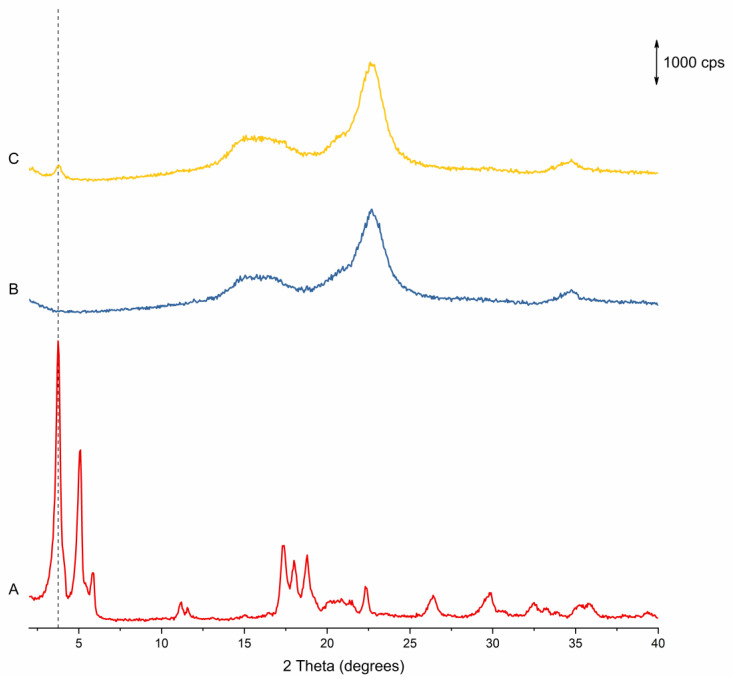
pXRD diffractograms for the materials used and produced in the fluidized bed coating study. (**A**) sodium ibuprofen (anhydrous), (**B**) uncoated MCC beads and (**C**) coated beads. The reference line is at 3.75 degrees (2θ).

**Figure 7 pharmaceutics-14-01058-f007:**
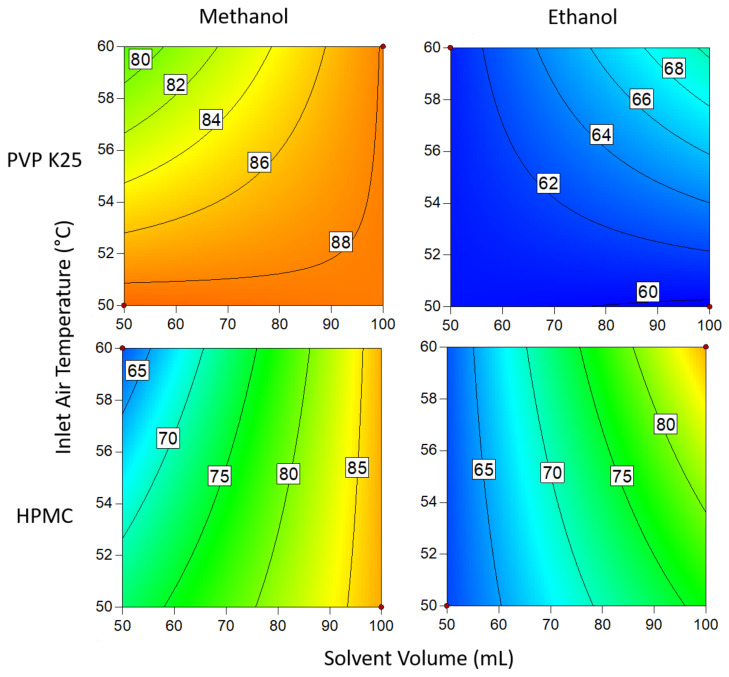
Drug loading efficiency when 1 g of binder is used. Numbers and tie lines correspond to points of equal DLE (%).

**Figure 8 pharmaceutics-14-01058-f008:**
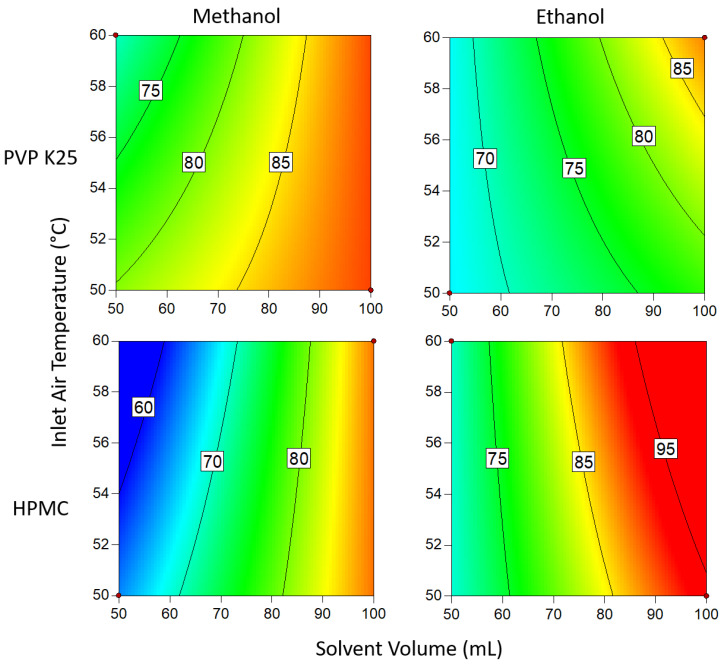
Drug loading efficiency when 3 g of binder is used. Numbers and tie lines correspond to points of equal DLE (%).

**Figure 9 pharmaceutics-14-01058-f009:**
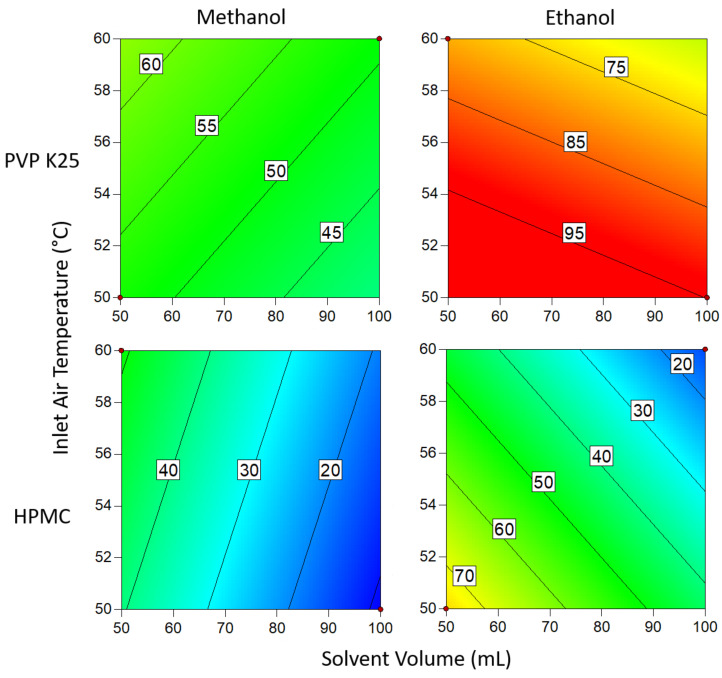
Degree of crystallinity when 1 g of binder is used. Numbers and tie lines correspond to points of equal DoC (%).

**Figure 10 pharmaceutics-14-01058-f010:**
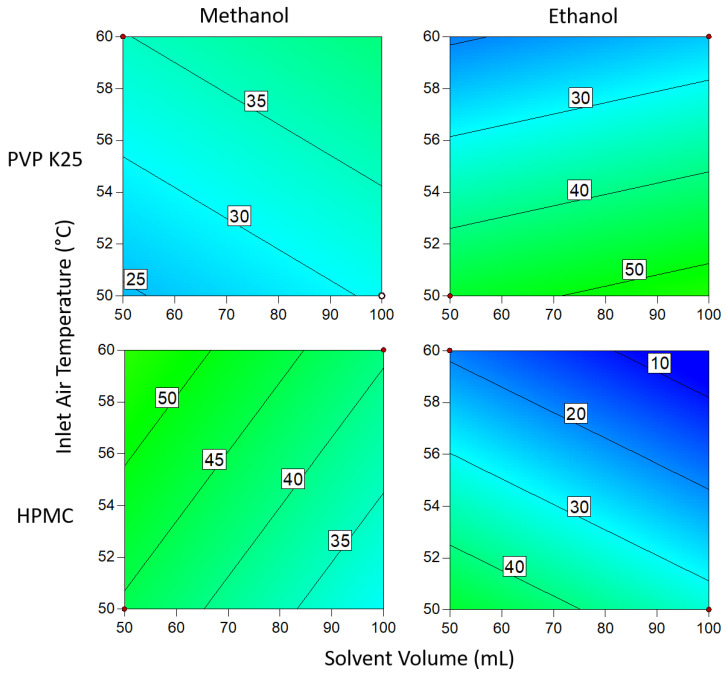
Degree of crystallinity when 3 g of binder is used. Numbers and tie lines correspond to points of equal DoC (%).

**Table 1 pharmaceutics-14-01058-t001:** Runs generated for the 2-level 5-factorial DoE study.

Run	Inlet Air Temperature (°C)	Binder Type	Binder Mass (g)	Solvent Volume (mL)	Solvent
1	60	HPMC	3	50	Ethanol
2	50	HPMC	3	100	Ethanol
3	60	HPMC	1	100	Ethanol
4	60	PVP	3	100	Ethanol
5	60	HPMC	3	100	Methanol
6	60	PVP	1	50	Ethanol
7	50	PVP	1	50	Methanol
8	50	HPMC	1	100	Methanol
9	50	PVP	1	100	Ethanol
10	50	PVP	3	100	Methanol
11	50	HPMC	1	50	Ethanol
12	50	PVP	3	50	Ethanol
13	60	PVP	3	50	Methanol
14	50	HPMC	3	50	Methanol
15	60	HPMC	1	50	Methanol
16	60	PVP	1	100	Methanol

**Table 2 pharmaceutics-14-01058-t002:** Results from the DoE study.

Run	Drug Loading Efficiency (%)	Degree of Crystallinity (%)
1	73	16
2	92	34
3	84	14
4	88	29
5	92	40
6	61	78
7	90	53
8	88	9
9	60	93
10	90	27
11	63	77
12	67	46
13	69	36
14	62	47
15	60	48
16	89	53

**Table 3 pharmaceutics-14-01058-t003:** Runs carried out in conjunction with the crystallizer.

Run	Aim	Inlet Air Temperature (°C)	Binder Type	Binder Mass (g)	Solvent Volume (mL)	Solvent
(i)	High DLE	60	HPMC	3	100	Ethanol
(ii)	High DoC	50	PVP	1	100	Ethanol

**Table 4 pharmaceutics-14-01058-t004:** Predicted values and actual results from carrying out the optimized spray coating runs using feed from the agitated bed crystallizer.

Run	Predicted DLE (%)	Actual DLE (%)	Predicted DoC (%)	Actual DoC (%)
(i)	105	94	8	8
(ii)	60	34	95	88

## Data Availability

Not applicable.

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
