# Peer review of "Integrated Purification and Formulation of an Active Pharmaceutical Ingredient via Agitated Bed Crystallization and Fluidized Bed Processing"

_pharmaceutics, 2022, doi:10.3390/pharmaceutics14051058_

Round 1

Reviewer 1 Report

Dear authors,

Presented work is thoroughly done and well presented.

I have few points to state that could contribute to work quility.  

Purification process and results connected to it should be separetly presented. Also SEM view could be beneficial or particle size uniformity. Solubility of formuleted drug could be affected so solubility and or dissolution studies would be benefitial.

Best of luck

Author Response

  1. The reviewer’s comment regarding the separate purification process are appreciated. A more in-depth discussion around this is being prepared as a separate publication. The present paper serves to demonstrate a possible application of this process and the advantages that might be conferred by it.
  2. SEM and particle sizing were considered for this work, however as a relatively small amount of material was being added to the surface of the spherical cores it was decided that this information would not have shown significant differences to the starting raw materials.
  3. The reviewer makes a valid point regarding the solubility of the drug and the dissolution performance. The purpose of this work was to develop new manufacturing processes and so this was outside the scope of the current paper, however this will be considered for the follow-on manuscript.

Reviewer 2 Report

The article entitled " Integrated Purification and Formulation of an Active Pharmaceutical Ingredient via Agitated Bed Crystallization and Fluidized Bed Processing" deals with the isolation-free confined suspension crystallization process fully integrated with a fluidized bed coater to enable isolation-free manufacturing of solid-state formulation of a model API, which is sodium ibuprofen.

The article is very well written and presented. The Introduction presents the state-of-the-art in the topic of large-scale API preparation(purification). The authors then presented the purification of a model drug, with a model impurity, after which, without the isolation of a solid API phase, a solid-state formulation is prepared. The Authors also used the DoE approach to the process parameters to control the drug loading efficiency and API crystallinity. The materials and methods part is written very clearly.

When it comes to the Results and Discussion part is presented very thoroughly, considering every aspect of the research described. The most favorable conditions of formulation of API are presented, considering the degree of crystallinity and drug loading efficiency. All parameters are considered, e.g., choosing of solvent type and volume, binder type, and amount and temperature of the process. According to the presented results, the conclusions are also clear.

One major issue I believe needs to be clearly explained by the Authors is the results concerning the degree of crystallinity, which, according to the description 2.6. should be quantified from the modulated DSC experiments, which is not explained in the Results and Discussion part.

Author Response

  1. The authors thank the reviewer for their comment regarding the degree of crystallinity. We have reiterated the method in the results and discussion, and this can found on Lines 312 to 314 of the revised manuscript. The results of degree of crystallinity as determined by modulated DSC are shown in section 3.1.1 (Table 2), section 3.1.4 (Figures 9 and 10) and section 3.2 (Tables 3 and 4).

Reviewer 3 Report

Good morning authors,

The paper submitted was very interesting and and presented novel work. It would appeal directly, and would be very useful to, the readership of the journal pharmaceutics. The design of the experiments and the delivery of the technology has been expertly carried out and reported. However there are a number of points that need to be addressed to further improve the manuscript and increase its impact. These largely centre on the analytical parts of the work. Once the detailed comments given below have been actioned or addressed the paper will be ready for publication.

1) line 22. In the abstract ibuprofen ethyl ester was stated as the chosen model impurity. However within the rest of the manuscript there was no comment concerning this particular form ibuprofen, especially in the methods and materials section. Apologies if I've missed a description of the ethyl ester in the manuscript, but in my reading I I couldn't find it. Thus please edit the manuscript accordingly to include a description of why the ethyl ester was used, and details about its purity et cetera need to go into the method section.

2) General point concerning the abstract. Apologies but I always like clear aim within an abstract, as it nicely connects the title and the conclusion of the work. Thus if it's possible please edit the abstract to clearly state the aim rather than implying it. 

3) The introduction was very well written and nicely presented the knowledge gap and the need for the work.

4) Excellent detail was given concerning the design of experiment, and the use of polymeric excipients. But a deeper rationale was required for the selection specifically of PVP K 25 and HPMC. This needs to be placed within the introduction or added to the discussion. Furthermore many of the readers will be familiar with the PVP classification K 25, but a little bit more detail concerning the molecular weight and molecular weight distribution of HPMC Pharma coat 606 was required. As there are lots of different grades of HPMC available.

5) Drug loading efficiency is one of the key dependent variables measured within the study. Thus it would be useful to know the limits of detection and quantification for the HPLC method applied, and also the validation of the approach to confirm that all of the drug had been removed from the beads. If it is not possible to retrospectively determine the LOD and LOQ an estimation would be sufficient, but more information concerning the validation of the extraction of the API is required.

6) The reported degree of crystallisation within the results section never quite gets to 100%, the highest is just over 80%. This may well be the true situation, however more detail is required in terms of the degree of crystallisation measurements using the DSC. This may be included in section 2.6 or even a critical evaluation within the discussion. Firstly what are the units of the heat of fusion for the API on the beads and the as received crystalline material. Is this joules per gram joules per mole? Next how would the presence of the bead effect the melting of the API. Small amounts of impurities affect both the onset temperature and and the enthalpy of melting for crystalline materials. The authors need to discuss the impact of any colligative effects. A simple experiment would be to make a physical mix of the bead and the crystalline API, to check that in the presence of the bead the melting profile of the crystalline drug does not alter from the expected enthalpy for the pure crystalline drug. Furthermore how does the reader know that the crystalline API is fully crystalline and of the correct polymorph? The sodium salt of ibuprofen has been investigated thoroughly by thermal analysis and so literature values of the expected enthalpy of fusion need to be compared to the measured results presented in your paper. The title of the paper concerns controlling crystallisation, so therefore more information concerning the validation of the method used to determine the degree of crystallinity is required. If there is some variance in the observed degree of crystallinity this will not lower the impact of the excellent work presented in the paper, it probably shows the difficulty in measuring small differences in crystallinity which is accepted throughout the field the pharmaceutical analysis.

7) In table 2 the loading efficiency and degree of crystallinity have been quoted to 2 decimal places, this seems to be a very high level of precision. Thus please provide a discussion of the errors and the statistics associated with recording these dependent properties. To put this in context the purity of the as received API is quoted to be above 99%, therefore the precision quoted for the degree of crystallinity and the drug loading is two orders of magnitude higher. For the HPLC assay in order to get to this level a reference standard should have also been run. Either provide the evidence or add a short discussion estimating and critically evaluating the precision of the assays.

8) A broad point concerning the excellent data presented in figures 8 and 9.  In the discussion there is no mention of the solubility of the API in either methanol or ethanol, would this make a contribution to the observed results? 

9) Other than the minor points listed above, the paper presents a really novel and interesting study, and should be published after these points have been addressed.

Author Response

  1. The reviewer makes a good point regarding the ibuprofen ethyl ester impurity. Discussion concerning why this particular impurity was used (structurally similar impurity which results in a realistic test system) can be found in section 3.2, lines 470-472. The impurity preparation has been added to the materials and methods section, lines 173-175.
  2. The aim has now been included in the abstract, lines 21-23.
  3. We thank the reviewer for their comment.
  4. HPMC 606 and PVP K25 were selected as binders as they have previously been used successfully as binders for granulating ibuprofen and MCC in a fluidized bed process. This has been included in section 3.1, lines 286-289. The molecular weights of both polymers have been included in the materials and methods section.
  5. At the time of the experiments it was not deemed to be important to determine the limit of detection of the HPLC method. A standard curve was prepared between 8.9 and 70.8 µg mL-1 and was validated weekly using an external standard. This limit of quantification has been included in the materials and method section, lines 249-250.
  6. The reviewer makes an interesting point regarding the degree of crystallinity of the API. The values measured by the DSC for the samples produced in the study were compared to a value found in the literature (all values were calculated in J/g). This has been included in section 2.6, lines 257-258. Secondly, the reviewer makes an acute observation concerning the influence of the other excipients on the melting profile of the drug and the polymorphic nature of the API. From the thermal data, there is a depression in the melting point of the drug relative to values quoted in the literature which is attributed to interactions between the polymer and API. Whilst this might suggest that the polymorph on the surface of the bead is different to that used as the reference material for the calculations (only metastable α and β forms have been previously reported along with the stable γ form usually reported), the alternative forms were previously produced by melt quenching the stable polymorph and when reheated, the melting events of both polymorphs were observed [1]. As these additional melting events were not observed for the coated beads in our study, this indicates that only the stable γ polymorph is present. Furthermore, a search of the CCDC indicates no polymorphs of sodium ibuprofen have been otherwise reported. Additionally, the PXRD for the samples consistently showed a small peak at 3.75° 2θ, again indicative of the γ form of the drug (the characteristic peaks of the α and β forms are expected at 5.2 and 5.7°, and 5.9° 2θ, respectively). This has been discussed in the manuscript in section 3.1.2, lines 353-358. The reviewers comments therefore bring about the interesting possibility of controlling the morphology of the API as part of the process which will be explored in future work.
  7. The level of precision of the data in Tables 2 and 4 has been reduced to that of the as-received API.
  8. We thank the reviewer for raising this point as it has added to the discussion. The contribution of the differences in solubility is now discussed in section 3.1.4, lines 426-428.
  9. We thank the reviewer for their comment.

Reviewer 4 Report

  1. Authors must mention the condition used to prepare the formulation.
  2. I donot agree with the authors research design. they have used different solvent and polymers. How they will get the optimized composition . Every solvent and polymer having the different property.
  3. How much is the desirability value of each factor.
  4. The objective need to be clear.
  5. What is the rationale to select ibuprofen as drug of choice

Author Response

  1. The conditions used to prepare the formulations can be found in Table 1 (runs used for the design of experiments study to develop the spray coating process) and Table 3 (runs carried out in conjunction with the crystalliser).
  2. Fluidised bed coating is a complex process with many factors affecting final product attributes. Polymer and solvent selection are key process variable in fluidised bed coating and so have been included in this study. The design of experiments approach taken here is commonly used with categoric variables (including different solvents and polymers). The effects of the polymer and solvent combinations of the drug loading efficiency and degree of crystallinity are discussed in sections 3.1.3 and 3.1.4, respectively. This information was used by the software to optimise the variable for each response in the combined crystallisation and coating runs.
  3. It is not possible to set desirability for factors during optimisation, however it is for possible for responses. The desirability for the desired response was set to 1.00 for the numerical optimisation.
  4. The objective is stated in the abstract, lines 21-23, and in the introduction lines, 148 to 156 of the revised manuscript. We are happy to expand on this in the abstract if required by the reviewer. The objective was to use a Design of Experiments (DoE) approach to develop a spray coating process that allows for the drug loading efficiency and the degree of crystalline material on the product to be tightly controlled. The spray coating process is fed with the product from a confined suspension-agitated bed suspension crystallization process, resulting in substantial process intensification, as purification and formulation are performed in two unit operations which can be adapted to run semi-continuously and integrated with existing continuous manufacturing processes.
  5. Ibuprofen is widely used as a model drug owing to its low solubility (though not relevant in this study), poor material properties (relatively low melting point, sticking on tabletting) and low cost.

Round 2

Reviewer 4 Report

Authors have revised the manuscript so recommended for acceptance.